# Adrenal Incidentalomas and Other Endocrine-Related Adenomas: How Much Does Cortisol Secretion Matter?

**DOI:** 10.3390/cancers15194735

**Published:** 2023-09-26

**Authors:** Aura D. Herrera-Martínez, Ángel Rebollo Román, Eider Pascual Corrales, Cindy Idrobo, Paola Parra Ramírez, Patricia Martín Rojas-Marcos, Cristina Robles Lázaro, Delia Lavinia Marginean, Marta Araujo-Castro

**Affiliations:** 1Endocrinology & Nutrition Department, Hospital Reina Sofia, 14004 Córdoba, Spain; angel.rebollo.sspa@juntadeandalucia.es (Á.R.R.); delia.marigean@juntadeandalucia.es (D.L.M.); 2Maimonides Institute for Biomedical Research of Cordoba (IMIBIC), 14004 Córdoba, Spain; 3Endocrinology & Nutrition Department, Hospital Universitario Ramón y Cajal, 28034 Madrid, Spaincindyidrobo@gmail.com (C.I.); 4Instituto de Investigación Biomédica Ramón y Cajal (IRYCIS), 28034 Madrid, Spain; 5Endocrinology & Nutrition Department, Hospital La Paz, 28034 Madrid, Spainpmrojasmarcos@salud.madrid.org (P.M.R.-M.); 6Endocrinology & Nutrition Department, Hospital de Salamanca, 37007 Salamanca, Spain

**Keywords:** cortisol, adrenal incidentaloma, endocrine-related adenomas, mortality

## Abstract

**Simple Summary:**

This article reflects a comprehensive analysis between cortisol secretion and the presence of other endocrine-related adenomas (specifically thyroid, parathyroid and pituitary). Cortisol has anti-inflammatory properties but has also been related to impaired cell proliferation and function. Specifically, its role in the presence of other benign lesions has not been described. For these reasons, we analyzed the prevalence of these other endocrine-related benign lesions in patients with nonfunctioning adrenal incidentalomas and with mild autonomous cortisol secretion. We observed that mild autonomous hypercortisolism does not affect the prevalence of other endocrine-related adenomas but is associated with increased metabolic comorbidities and mortality in these patients.

**Abstract:**

**Background:** Adrenal incidentalomas (AI) are frequent findings in clinical practice. About 40% of AIs are associated with hypercortisolism of variable severity. Although mild autonomous cortisol secretion (MACS) has been associated with the impaired clinical outcome of several diseases, its effect on the development of benign neoplasms is unknown. **Aim**: To compare the prevalence of adenomas (thyroid, parathyroid, pituitary and other locations) in patients with nonfunctioning AIs (NFAIs) and MACS. **Methods**: A multicenter, retrospective study of patients with AIs evaluated in four tertiary hospitals was performed. **Results**: A total of 923 patients were included. Most patients were male (53.6%), with a mean age at diagnosis of 62.4 ± 11.13 years; 21.7% presented with bilateral AIs. MACS was observed in 29.9% (n = 276) of patients, while 69.9% (n = 647) were NFAIs. Adenomas in locations other than the adrenal gland were observed in 36% of the studied population, with a similar distribution in patients with MACS and NFAIs (33% vs. 32%; *p* > 0.05). There were no statistically significant differences in the prevalence of pituitary, thyroid, parathyroid or other endocrine-related adenomas between both groups, but the prevalence of metabolic comorbidities and mortality was increased in patients with MACS, specifically in patients with thyroid and other endocrine-related adenomas (*p* < 0.05). **Conclusions**: Adenomas in locations other than the adrenal glands occur in one third of patients with AIs. Mild autonomous hypercortisolism does not affect the prevalence of other endocrine-related adenomas but is associated with increased metabolic comorbidities and mortality, especially in patients with thyroid adenomas and adenomas in other locations.

## 1. Introduction

The most common endocrine-related incidentalomas are located in the adrenal, thyroid and pituitary glands [1,2]. Due to improvement in radiological techniques and early detection of diseases, the increase in prevalence of endocrine-related incidentalomas is expected to continue [3]. Specifically, adrenal incidentalomas are mass lesions detected on imaging performed for another clinical reason different to suspicion of adrenal disease [4].

It is known that multiple endocrine neoplasia syndrome type 1 (MEN1) is related to the presence of adenomas in multiple locations, but the presence of endocrine-related adenomas in patients without MEN1 syndrome has not been reported, probably due to their benign behavior or lower prevalence [5]. In contrast, several case reports in the literature only reflect the presence of endocrine-related adenomas along with malignant neoplasms or with unusual hormone secretion [6,7,8,9,10,11].

Adrenal incidentalomas are reported in approximately 4.2% of imaging studies, and their prevalence reaches about 6–7% in autopsies. Their prevalence is age-related; specifically, it is about 0.2% in patients younger than 30 years and increases up to 7% in patients older than 70 years [12]. Despite an adrenal incidentaloma being usually asymptomatic, an appropriate hormonal evaluation is necessary, since up to 55% can present with excess hormone secretion [2,4]. Specifically, mild autonomous cortisol secretion (MACS) is the most common hormonal finding, being observed in approximately 30% of adrenal incidentalomas [13]. It has been previously named subclinical hypercortisolism due to the presence of excess biochemical cortisol in the absence of classic clinical features of Cushing syndrome [2,4], but it has been related to the presence of metabolic comorbidities or impaired control [1].

Cortisol secretion has been previously associated with impaired cell proliferation, suggesting that hypercortisolism could be directly related to increased prevalence of benign and malignant lesions [14,15,16]. Other studies, in contrast, have suggested that cortisol-related systemic alterations (in blood vessels and circulation) could alter the histological characteristics of different glands, contributing to the presence of adenomas in different locations [17]. Furthermore, the dose-dependent effects of cortisol on cell proliferation have been reported in vitro: specifically, cortisol may increase mitosis, cell migration and cytokines release, which increases tumor proliferation in head and neck cancers [14]. Additionally, chronic cortisol secretion produce functional epigenomic changes, which can induce specific cell phenotypes that have been related to several diseases including cancer [18]. Furthermore, its effects might be tissue-specific; for example, an antiproliferative effect has been described in bone cells, which affects both healthy cells (in consequence producing osteoporosis) [19,20] and osteosarcoma cells, avoiding tumor spread and metastasis [21].

In addition to these controversial results, cortisol secretion has been associated with increased cardiovascular risk. Specifically, the prevalence of cardiovascular outcomes is three times higher in patients with MACS than in patients with NFAIs [22]. Current European clinical guidelines suggest individualizing surgical management depending on cortisol levels and the clinical evolution of comorbidities [2,4], but the presence of other lesions, including benign tumors such as endocrine-related adenomas, is not considered. In this context, we aimed to determine the prevalence of other endocrine-related adenomas in patients with adrenal incidentalomas and to compare the clinical characteristics, including metabolic comorbidities and mortality, depending on the presence of MACS.

## 2. Materials and Methods

### 2.1. Patients

This study was approved by the Ethics Committee of the Ramón y Cajal University Hospital (Registration number 3702, on 23 May 2019, Madrid, Spain). Additionally, it was conducted in accordance with the Declaration of Helsinki and according to national and international guidelines. This is a multicenter retrospective study, wherein patients with adrenal incidentalomas from four Spanish Hospitals were included (Reina Sofía University Hospital (Córdoba), Ramón y Cajal University Hospital (Madrid), La Paz University Hospital (Madrid) and Salamanca University Hospital (Salamanca)). Nine hundred twenty-three (n = 923) patients with adrenal incidentalomas (evaluated during a twenty-year period (2002–2023)) were included. Specifically, only NFAIs and MACS were considered in this study; adrenal lesions with other hormone secretion were excluded. Clinical records were used to collect full medical histories. All patients were managed according to available guidelines and recommendations [2,4]. The prevalence of endocrine-related adenomas in other locations than the adrenal gland was evaluated; additional information about comorbidities and mortality was also collected. Clinical and/or surgical management of the adenomas was performed by the corresponding specialists according to the specific clinical guidelines. The definitions of MACS-related comorbidities have been previously described [13].

### 2.2. Imaging Techniques

Adrenal incidentalomas were characterized using unenhanced computed tomography (CT) scans or magnetic resonance imaging (MRI) as benign lesions with a tumor density equal to or less than 10 HU on unenhanced CT [2]. Pituitary, parathyroid and thyroid adenomas were evaluated separately. Adenomas in other locations were clustered in a single group named “others”. The identification of endocrine-related adenomas was performed using different imaging techniques, which were performed when necessary: MRI for pituitary adenomas (only non-functioning lesions were included in this study); neck ultrasound and scintigraphy were used for identifying parathyroid adenomas; and thyroid nodules were evaluated using neck ultrasound, where only nodules with low suspicion of malignancy (using Tirads score [23]), a negative fine-needle aspiration punction for malignancy or a histologically confirmed benign thyroid tumor were included in this study. Adenomas in other locations (skin and soft tissue) were diagnosed using ultrasound or axial tomography.

### 2.3. Hormonal Data and Definitions

A hormonal study of the adrenal incidentaloma was performed as indicated by current clinical guidelines [2,4]. Specifically, the biochemical analysis included: serum baseline morning cortisol (expressed in mcg/dl), adrenocorticotropic hormone ((ACTH), expressed in ng/dL; this parameter was available in 569 patients), dehydroepiandrostendione sulphate ((DHEAS), expressed in ng/dl; this parameter was available in 504 patients), 24 h urinary free cortisol ((UFC), expressed in mcg/24 h; UFC was available in 378 patients), and 1 mg dexamethasone suppression test ((DST), expressed in mcg/dL; which was available in all patients). Only patients with nonfunctioning adrenal incidentalomas (1 mg DST < 1.8 mcg/dL) and MACS adenomas (1 mg DST 1.8–5 mcg/dL) were included.

### 2.4. Statistical Analysis

Continuous variables were expressed as medians with interquartile ranges, and categorical variables were described as proportions. For missing data and specific group analysis, the absolute number is also expressed in brackets.

Between-group comparisons were analyzed by the U Mann–Whitney test (nonparametric data). Paired analysis was performed by Student’s *t* test (parametric data) or Wilcoxon test (nonparametric data). Chi-squared test was used to compare categorical data. Statistical analyses were performed using SPSS statistical software version 20 and Graph Pad Prism version 9. Then, *p* values < 0.05 were considered statistically significant.

## 3. Results

### 3.1. Baseline Characteristics of MACS and NFAI

Nine hundred twenty-three patients were included, with 53.6% males and 62 years median age at diagnosis of the adrenal incidentaloma. About 40% of the patients were former smokers and 27% active smokers. NFAIs occurred in 70% of patients, while MACS was observed in 30% of the cohort (n = 276). Additionally, 21.7% were presented with bilateral adrenal incidentalomas. There were no statistically significant differences between both groups when the size of the AI was compared.

Patients with MACS were diagnosed at a slightly older age than patients with NFAIs (64 ± 10.32 vs. 62 ± 11.04, *p* < 0.05). Regarding metabolic comorbidities, hypertension was significantly more prevalent in patients with MACS (62% vs. 51%, *p* < 0.001), whereas no statistically significant differences were detected in other comorbidities, including diabetes, obesity, cardiovascular or cerebrovascular complications. Mortality was increased in patients with MACS compared with patients with NFAIs (4.3% vs. 1.2%, *p* < 0.01). Detailed characteristics of both groups and the whole cohort are depicted in Table 1.

### 3.2. Biochemical Profile of Patients with and without Other Endocrine-Related Adenomas

In NFAIs, serum ACTH levels were increased in patients with adenomas when compared with patients without adenomas. Additionally, cortisol after midnight dexamethasone was also increased in these patients (Figure 1A). In patients with MACS, 24 h urinary cortisol excretion was lower in patients with adenomas compared with patients without adenomas (Figure 1B). No other significant changes in baseline cortisol or serum DHEAS were observed.

### 3.3. Adrenal Incidentalomas and Other Endocrine-Related Adenomas

In the whole cohort, 36% presented an adenoma in a different location than the adrenal gland, with similar prevalence in patients with NFAIs and MACS (32% and 33%, *p* = 0.41). Pituitary, parathyroid and thyroid adenomas were evaluated separately. Among them, thyroid adenomas were the most prevalent (24.4%), with no significant differences in the prevalence in patients with NFAIs and MACS (25.2% vs. 23.9%, *p* = 0.38). Skin and soft-tissue adenomas were clustered in a single group named other adenomas; their prevalence reached 33.4% (31.8% in NFAIs vs. 33% in MACS, *p* = 0.38). Age-adjusted cortisol secretion was not associated with the prevalence of adenomas (OR 1.02, CI 0.75–1.38) (Table 2), but it was associated with mortality (OR 3.30, CI 1.32–8.23) (Table 2).

### 3.4. Pituitary Adenomas

A total of 38 patients presented with a pituitary adenoma (4.1%). Despite including only patients with nonfunctioning pituitary adenomas in this study, the prevalence of diabetes tended to be increased in patients with MACS when compared with patients with NFAIs (75% vs. 4.4%; *p* = 0.08; Table 3). There were no statistically significant differences between groups when age, sex other metabolic comorbidities and mortality were compared. Age-adjusted cortisol secretion was not associated with the presence of pituitary adenomas (OR 1.34, CI 0.39–4.46; Table 2).

### 3.5. Thyroid Adenomas

Two hundred twenty-nine patients presented with thyroid adenomas. There were no differences in the age and sex of patients with thyroid nodules and NFAIs or MACS. Nevertheless, in this group of patients, metabolic comorbidities were more prevalent in patients with MACS than in patients with NFAIs, specifically hypertension (66.2% vs. 49.1%, *p* = 0.01), diabetes (31% vs. 19%, *p* = 0.04) and cerebrovascular complications (18.5% vs. 9.1%; *p* = 0.04). Mortality was higher in patients with MACS and thyroid adenomas (6.2% vs. 0%; *p* < 0.01; Table 4). Age-adjusted cortisol secretion was not associated with the presence of thyroid adenomas (OR 0.93 CI 0.66–1.29; Table 2).

### 3.6. Parathyroid Adenomas

Parathyroid adenomas were detected in twenty-three patients. Patients with parathyroid adenomas and NFAIs or MACS did not present differences when age, sex, metabolic comorbidities and mortality were compared (Table 5). Age-adjusted cortisol secretion was not associated with the presence of parathyroid adenomas (OR 0.99, CI 0.40–2.46, Table 2).

### 3.7. Adenomas in Other Locations

Adenomas in other locations were observed in 297 patients. In this group, some metabolic complications were increased in patients with MACS than in patients with NFAIs: specifically, hypertension (65.6% vs. 51%; *p* = 0.01), diabetes (31.1%ª vs. 19.4%, *p* = 0.02) and cerebrovascular complications (18.5% vs. 9.2%, *p* = 0.04). There were no statistically significant differences in age, sex and other metabolic comorbidities (Table 6). Also in this group, mortality was significantly increased in patients with MACS (4.4%) than in patients with NFAIs (0.5%; *p* < 0.05). Age-adjusted cortisol secretion was not associated with the presence of adenomas in other locations (OR 1.04, CI 0.76–1.41; Table 2).

### 3.8. Metabolic Comorbidities in Patients with MACS and Adenomas

Metabolic comorbidities were separately analyzed in patients with MACS. The prevalence of cerebrovascular complications was increased in patients with MACS and other endocrine-related adenomas compared with patients without adenomas (18.9% vs. 9.4%, *p* < 0.02). In contrast, there were no significant differences in the prevalence of hypertension, diabetes, dyslipidemia, cardiovascular complications, obesity and mortality in patients with and without adenomas (Table 7).

## 4. Discussion

Current clinical practice entails the improved performance of numerous imaging techniques, resulting in increased diagnoses of incidental lesions, especially endocrine-related adenomas. In this context, adrenal incidentalomas represent a particularly important entity due to the possible presence of cortisol secretion and its related effects on morbidity and mortality. The prevalence of adrenal incidentalomas varies depending on age and the studied population. Imaging-based studies have reported an age-related prevalence, specifically in the range of 4–7% among people older than 40 years and up to 5–10% in patients older than 70 years [24]. Autopsy series report prevalence of about 7% [25]. Due to their relatively high prevalence in imaging studies, adrenal incidentalomas are a frequent consultation for endocrinologists.

In this context, our study aimed to report some clinical characteristics of patients with NFAIs and MACS, especially focusing on the prevalence of other endocrine-related adenomas. To the best of our knowledge, this is the first specific report of endocrine-related adenomas in a cohort of patients with adrenal incidentalomas.

As in our study, earlier reports have described a median age of adrenal incidentaloma diagnosis of 62 years old [26]. Nevertheless, we found that adrenal incidentalomas were more prevalent in men, in contrast to previous reports, in which a slightly increased prevalence in women (55%) has been reported [26]. Additionally, we observed a higher prevalence of MACS (30%) compared with that of other studies (20%) [27,28] but similar to other population-based reports (37%) [26]. Additionally, an increased number of bilateral lesions was observed in our study (21.7%) compared with that in earlier reports in the literature (4.5%) [26]. Differences across studies may be justified due to the different definitions of MACS employed and the different designs of the studies.

A recent metanalysis described a thyroid nodule prevalence of 24.83% in the general population, and the prevalence has increased during the last ten years, especially in women (36.5%1) compared with men (23.47%) [29]. Furthermore, in the metanalysis, obesity was correlated with the prevalence of thyroid nodules [29]. This finding was not observed in our cohort.

In this line, previous studies have described a significantly increased prevalence of thyroid nodules in patients with Cushing syndrome [30], especially in Cushing disease (60%) [30] compared with adrenal Cushing syndrome (25–30%) [31], but no increase has been observed in patients with NFAIs [30]. Based on this, it has been hypothesized that an unknown factor may affect both adrenal and thyroid glands when cortisol secretion is increased [32]. In this study, we did not evaluate patients with adrenal Cushing syndrome, but in our cohort, we did not observe significant differences in the prevalence of thyroid nodules in patients with MACS of NFAIs.

Remarkably, we observed increased metabolic comorbidities in patients with MACS, thyroid adenomas and adenomas in other locations. Similarly, previous studies have described an increased prevalence of adrenal incidentalomas in elderly patients, which has been associated with increased arteriopathy of adrenal capsule arteries [17]. Additionally, a higher prevalence of adrenal incidentalomas has been also reported in patients with hyperthyroidism, type 2 diabetes and hypertension [25,32,33]. Moreover, MACS has been associated with other metabolic comorbidities including insulin resistance, obesity and metabolic syndrome [34,35,36]. These relations are clearly observed in this study, specifically in the cohort of patients with thyroid adenomas and adenomas in other locations. The specific underlying mechanism is unknown, but altered inflammation, cytokine release, decreased plasma nitrate/nitrite concentrations and altered microcirculation have been proposed [37,38,39,40]. Furthermore, it has been also suggested that chronic cortisol excess is associated with impaired brain blood flow, which might affect the brain microenvironment and as a consequence produce impaired cognitive function [41]. In our cohort, patients with thyroid adenomas and adenomas in other locations presented with increased prevalence of cerebrovascular complications, but other clinical conditions such as dementia were not considered. Furthermore, MACS was independently associated with increased mortality as previously reported [42]. 

Regarding pituitary adenomas, these are a diverse group of tumors arising from the pituitary gland; specifically, an overall estimated prevalence of pituitary adenomas of 16.7% has been reported (14.4% in autopsy studies and 22.5% in imaging studies) [43]. Historically, these tumors have been classified according to size and divided into microadenomas (maximal tumor diameter < 1 cm) and macroadenomas (maximal tumor diameter ≥ 1 cm). Due to the small size of many pituitary lesions, the fact that many are nonfunctioning pituitary adenomas or that they might present with a very mild hormone secretion, it is a challenge to accurately measure the real prevalence of pituitary adenomas in the general population [43]. Chrisoulidou et al. described a prevalence of 8% of pituitary adenomas in patients with adrenal incidentalomas [44]. In contrast, the prevalence rate in our cohort was almost half (4.1%). Similarly, MACS was not associated with the presence of pituitary adenomas [44]. Despite this, it has been described that almost 23% of patients with adrenal incidentaloma present an alteration in the pituitary gland morphology, suggesting that an altered mechanism could affect cell proliferation on both glands.

A parathyroid adenoma is a benign tumor of the parathyroid gland that generally causes hyperparathyroidism. It is usually diagnosed when patients present with hypercalcemia; specifically, a single parathyroid adenoma is responsible for 80–85% of hyperparathyroidism, double adenomas are observed in 4–5% of cases and parathyroid hyperplasia in 10–12% of patients. Additionally, parathyroid carcinomas are very rare causes of hyperparathyroidism and account for less than 1% of the disease cause [45,46].

We did not find in the literature specific reports about parathyroid adenomas and adrenal incidentalomas apart from studies in patients with MEN1 or MEN1-like syndromes. Although parathyroid adenomas are not uncommon, these lesions can be frequently missed due to their small size [47]. As previously described, hypercalcemia is frequently the leading sign for diagnosing these adenomas and is associated with several clinical outcomes, including increased cardiovascular risk and mortality. Specifically, hyperparathyroidism is associated with a higher incidence of hypertension, left ventricular hypertrophy, heart failure, cardiac arrhythmias and valvular calcific disease, which may contribute to higher cardiac morbidity and mortality [48]. In our cohort we did not observe an increased prevalence of metabolic comorbidities or mortality, probably due to an appropriate, early clinical and surgical management of the primary hyperparathyroidism.

This study has some limitations. First, the retrospective nature of the study is accompanied by an intrinsic risk of bias and missing data. Additionally, we did not include patients with overt cortisol secretion, which could have allowed us to compare differences across different degrees of cortisol secretion. We did not specifically analyze hormone levels in pituitary or parathyroid adenomas. Finally, only endocrine-related adenomas were included in the analysis, and other benign lesions were not included in the analysis. In contrast, the most important strength of this study is the large number of well-characterized patients with adrenal incidentalomas and the novelty of the findings.

## 5. Conclusions

Taken together, our results reveal that mild autonomous cortisol secretion in patients with adrenal incidentalomas does not affect the prevalence of adenomas in other locations, but metabolic comorbidities might be increased in patients with thyroid adenomas and adenomas in other locations. Additionally, mild autonomous cortisol secretion was significantly associated with increased mortality in patients with and without other endocrine-related adenomas. Additional studies in larger cohorts and specific analyses of patients with overt Cushing syndrome should be performed in order to validate and enlarge the results of this study.

## Figures and Tables

**Figure 1 cancers-15-04735-f001:**
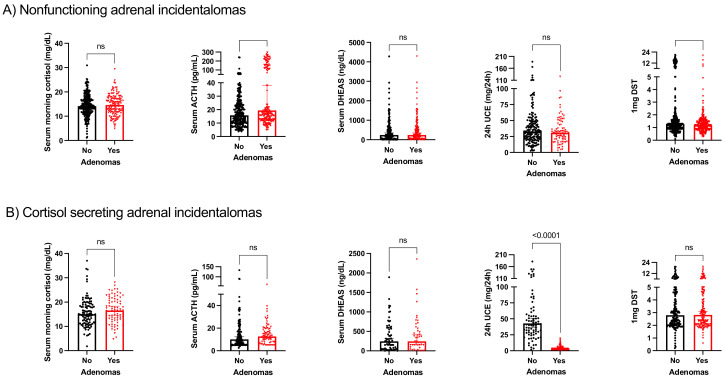
Comparison of hormone profile in patients with and without adenomas in (**A**) nonfunctioning adrenal incidentalomas; (**B**) cortisol-secreting adrenal incidentalomas. ns: no significance.

**Table 1 cancers-15-04735-t001:** General characteristics of all included patients.

Characteristic	All Patients (n = 923)	Nonfunctioning AI (n = 647)	MACS (n = 276)	*p*
Age at diagnosis (years)	62.4 ± 11.12	62.15 ± 11.04	64.12 ± 10.32	0.02
Sex (%male/%female)	53.6/46.4	52.9/47.1	55.3/44.7	0.28
Adenoma size	22.3 ± 3.5	20.3 ± 3.8	23.1 ± 2.0	0.57
**Tobacco exposure**				
Active smoker (% and n)	27.4 (52/190)	24.6 (34/138)	34.6 (18/52)	0.28
Former smoker (% and n)	39.9 (215/552)	40.1 (150/374)	36.5 (65/178)	0.11
**Complications**				
Hypertension (% and n)	54.7 (502/918)	51.4 (331/644)	62.4(171/274)	0.001
Diabetes (% and n)	24.5 (224/913)	23.3 (149/639)	27.4 (75/274)	0.11
Dyslipidemia (% and n)	48.2 (440/913)	47.3 (303/640)	50.2 (137/273)	0.23
Cardiovascular complications (% and n)	2 (18/914)	1.9 (12/640)	2.2 (6/274)	0.46
Cerebrovascular complications (% and n)	10.9 (99/910)	10.2 (65/640)	12.6 (34/270)	0.16
Obesity (% and n)	40.5 (293/723)	41.9 (209/499)	37.5 (84/224)	0.15
**Adenomas in other locations**	36 (298/827)	32 (207/647)	33 (91/276)	0.41
**Location**				
Pituitary (% and n)	4.1 (38/935)	1.1 (7/647)	1.4 (4/276)	0.42
Thyroid (% and n)	24.4 (229/935)	25.2 (163/647)	23.9 (66/276)	0.38
Parathyroid (% and n)	4.3 (40/935)	2.5 (16/647)	2.5 (7/276)	0.54
Others (% and n)	33.4 (297/889)	31.8 (206/647)	33 (91/276)	0.38
**Mortality** (% and n)	2.2 (20/923)	1.2 (8/647)	4.3 (12/276)	0.005

**Legend:** MACS: Mild autonomous cortisol secretion; AI: adrenal incidentaloma.

**Table 2 cancers-15-04735-t002:** Multivariate logistic regression for the presence of endocrine-related adenomas in patients with autonomous cortisol-secreting AI adjusted by age.

Variable		OR	CI	*p*
**Adenoma location**	All types of adenomas	1.02	0.75–1.38	0.89
	Pituitary	1.34	0.38–4.46	0.64
	Thyroid	0.91	0.65–1.28	0.76
	Parathyroid	0.99	0.40–2.45	0.98
	Other	1.03	0.76–1.40	0.82
**Mortality**		3.30	1.32–8.23	0.01

**Table 3 cancers-15-04735-t003:** Characteristics of patients with pituitary adenomas and adrenal incidentalomas.

Characteristic	All Patients (n = 11)	Nonfunctioning AI (n = 7)	MACS (n = 4)	*p*
Sex (male/female)	54.5/45.5	71.4/28.6	25/75	0.19
Age at diagnosis of AI	64.5 ± 9.20	65.6 ± 3.69	62.5 ± 4.56	0.78
**Tobacco exposure**				
Active smoker	0	0	0	-
Former smoker	22.2 (4/9)	20 (1/5)	25 (3/4)	0.72
**Complications**				
Hypertension	63.6 (7/11)	57.1 (4/7)	75 (3/4)	0.53
Diabetes	36.4 (4/11)	14.3 (1/7)	75 (3/4)	0.08
Dyslipidemia	27.3 (3/11)	28.6 (2/7)	25 (1/4)	0.72
Cardiovascular complications	0	0	0	-
Cerebrovascular complications	9.1 (1/11)	0	25 (1/4)	0.36
Obesity	71.4 (7/7)	50 (2/4)	100 (3/3)	0.28
Mortality	0	0	0	-

**Legend:** MACS: Mild autonomous cortisol secretion; AI: adrenal incidentaloma.

**Table 4 cancers-15-04735-t004:** Characteristics of patients with thyroid adenomas and adrenal incidentaloma.

Characteristic	All Patients (n = 229)	Nonfunctioning AI (n = 163)	MACS (n = 66)	*p*
Sex (male/female)	55/45	55.2/44.8	54.5/44.5	0.52
Age at diagnosis of AI	63.5 ± 10.80	63.1 ± 10.70	64.57 ± 10.90	0.39
**Tobacco exposure**				
Active smoker	9.1 (1/11)	0	33.3 (1/3)	0.27
Former smoker	40.6 (52/128)	43.7 (38/87)	34.1 (14/41)	0.20
**Complications**				
Hypertension	53.9 (123/228)	49.1 (80/163)	66.2 (43/65)	0.01
Diabetes	22.4 (51/228)	19.0 (31/163)	31.01 (20/65)	0.04
Dyslipidemia	50 (114/228)	46.6 (75/163)	58.5 (38/65)	0.07
Cardiovascular complications	0.9 (2/228)	0.6 (1/163)	1.5/1/65)	0.49
Cerebrovascular complications	11.8 (27/228)	9.2/15/63)	18.5 (12/65)	0.04
Obesity	43 (74/172)	47 (55/117)	34.5 (19/55)	0.08
Mortality	1.7 (4/229)	0	6.2 (4/66)	0.006

**Legend**: MACS: Mild autonomous cortisol secretion; AI: adrenal incidentaloma.

**Table 5 cancers-15-04735-t005:** Characteristics of patients with parathyroid adenomas and adrenal incidentaloma.

Characteristic	All Patients (n = 23)	Nonfunctioning AI (n = 16)	MACS (n = 7)	*p*
Sex (male/female)	69.6/30.4	68.8/31.3	71.4/28.6	0.64
Age at diagnosis of AI	66 ± 11.28	67.9 ± 10.23	61.6 ± 12.12	0.24
**Tobacco exposure**				
Active smoker	0	0	0	-
Former smoker	28.6 (4/14)	33.3 (4/12)	0	0.49
**Complications**				
Hypertension	60.9 (14/23)	56.3 (9/16)	71.4 (5/7)	0.42
Diabetes	26.1 (6/23)	25 (4/16)	2.6 (2/7)	0.61
Dyslipidemia	52.2 (12/23)	56.3 (9/16)	42.9 (3/7)	0.44
Cardiovascular complications	0	0	0	-
Cerebrovascular complications	4.3 (1/23)	0	14.3 (1/7)	0.30
Obesity	42.1 (8/19)	50 (7/14)	20 (1/5)	0.27
Mortality	0	0	0	-

**Legend**: MACS: Autonomous cortisol secretion; AI: adrenal incidentaloma.

**Table 6 cancers-15-04735-t006:** Characteristics of patients with other adenomas and adrenal incidentaloma.

Characteristic	All Patients (n = 297)	Nonfunctioning AI (n = 206)	MACS (n = 91)	*p*
Sex (male/female)	69.6/30.4	55.8/44.2	57.1/42.9	0.46
Age at diagnosis of AI	63.6 ± 10.11	63.5 ± 10.20	63.9 ± 11.01	0.92
**Tobacco exposure**				
Active smoker	6.3 (1/16)	0	16.7 (1/16)	0.37
Former smoker	38.7 (67/173)	41.6 (47/113)	33.3 (20/60)	0.42
**Complications**				
Hypertension	55.4 (164/296)	51 (105/206)	65.6 (59/90)	0.01
Diabetes	23 (68/296)	19.4 (40/26)	31.1 (28/90)	0.02
Dyslipidemia	52 (154/296)	50.5 (104/206)	55.6 (50/90)	0.24
Cardiovascular complications	0.7 (2/296)	0.5 (1/206)	1.1 (1/90)	0.51
Cerebrovascular complications	12.2 (36/296)	9.2 (19/206)	18.9 (17/90)	0.01
Obesity	42.3 (96/227)	45.4 (69/152)	36 (27/75)	0.11
Mortality	1.7 (5/297)	0.5 (1/206)	4.4 (4/91)	0.03

**Legend**: MACS: Mild autonomous cortisol secretion; AI: adrenal incidentaloma.

**Table 7 cancers-15-04735-t007:** Characteristics of patients with MACS and adenoma in other locations than the adrenal gland.

Characteristic	No Adenoma (n = 185)	Adenoma (n = 91)	*p*
Sex (male/female)	54.3/45.7	57.1/42.9	0.37
Age at diagnosis of AI	64.2 ± 11.41	63.9 ± 11.01	0.92
**Tobacco exposure**			
Active smoker	37 (17/46)	16.1 (1/6)	0.31
Former smoker	38.1 (45/118)	33.3 (20/60)	0.32
**Complications**			
Hypertension	60.9 (112/184)	65.6 (59/90)	0.27
Diabetes	25.5 (47/184)	31.1 (28/90)	0.20
Dyslipidemia	47.5 (87/183)	55.6 (50/90)	0.13
Cardiovascular complications	2.7 (5/184)	1.1 (1/90)	0.35
Cerebrovascular complications	9.4 (17/180)	18.9 (17/90)	0.02
Obesity	38.3 (57/149)	36 (27/75)	0.43
Mortality	4.3 (8/185)	4.4 (4/91)	0.60

## Data Availability

All data generated or analyzed during this study are included in this article. Further enquiries can be directed to the corresponding authors.

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
