# Peer review of "Adrenal Incidentalomas and Other Endocrine-Related Adenomas: How Much Does Cortisol Secretion Matter?"

_cancers, 2023, doi:10.3390/cancers15194735_

Round 1

Reviewer 1 Report

The authors present an interesting study that, although not novel, expands on existing clinical observational data. 

General points to consider:

1. Grammatical inconsistencies should be improved upon throughout.

2. In line 102/103, the authors incorrectly list 935 patients in italics as opposed to 923.

3. In general, a significant proportion of adrenal incidentalomas are associated with hyperaldosteronism. Given the significantly increased incidence of hypertension in some sub-groups analyzed, were clinical records probed for plasma renin activity/plasma aldosterone concentrations? Can the authors please comment on this.

4. The authors' conclusions are appropriate and corroborate the data presented. 

The language quality can be improved, specifically grammatical errors that undermine the coherency of the data being presented. I suggest the authors proof-read the manuscript once more, focusing on this aspect. 

Author Response

We sincerely thank the Reviewer for the constructive comments, which we found very helpful towards improving the quality of our study. Accordingly, specific changes have been made in the manuscript, based on these comments, as it is described in detail below in a point-by-point description of the changes introduced, and on how Reviewer’s concerns were addressed. Changes in the manuscript are indicated in red

REVIEWER 1

The authors present an interesting study that, although not novel, expands on existing clinical observational data. 

General points to consider:

  1. Grammatical inconsistencies should be improved upon throughout.

Authors: The revised version of our manuscript has been reviewed by a native English speaker in order to improve its quality and correct grammatical inconsistencies

  1. In line 102/103, the authors incorrectly list 935 patients in italics as opposed to 923.

Authors: We thank the reviewer for pointing this out, this inconsistence has been corrected

  1. In general, a significant proportion of adrenal incidentalomas are associated with hyperaldosteronism. Given the significantly increased incidence of hypertension in some sub-groups analyzed, were clinical records probed for plasma renin activity/plasma aldosterone concentrations? Can the authors please comment on this.

Authors: We thank the reviewer for this comment, only non-functioning adrenal incidentalomas and (mild) cortisol secreting lesions were included in this study. This information has been included in the revised version of the manuscript since it might bring into confusion

  1. The authors' conclusions are appropriate and corroborate the data presented. 

Authors: We thank the reviewer for this comment.

Reviewer 2 Report

Please see the attached review.

Moderate editing of English language required

Author Response

We sincerely thank the Reviewer for the constructive comments, which we found very helpful towards improving the quality of our study. Accordingly, specific changes have been made in the manuscript, based on these comments, as it is described in detail below in a point-by-point description of the changes introduced, and on how Reviewer’s concerns were addressed. Changes in the manuscript are indicated in red

REVIEWER 2

 Thank you for the opportunity to review the manuscript “Adrenal incidentalomas and other endocrine-related adenomas: How much does cortisol secretion matter?” by Aura D. Herrera-Martínez et al. It refers to an important issue of comorbidities in case of adrenal incidentalomas that are frequent imaging findings. I find the paper novel and interesting.

However, there are some issues that need to be improved.

  1. Please specify which years where included in the analysis.

Authors: Patients were evaluated during 2002-2023. This information has been included in the revised version of the manuscript

  1. Have all the patients had all scans (pituitary MRI, thyroid US, other US/CT)?

Authors: Image studies was performed only when necessary. Current guidelines do not recommend regular pituitary MRI, thyroid US or other explorations in patients with adrenal lesions, additionally, hereditary syndromes were not included, since adrenal lesions in these patients are not considered adrenal incidentalomas. For avoiding confusion, this information has been explicitly included in the revised version of the manuscript

  1. The information that pituitary and parathyroid adenomas were not evaluated if they are functioning should be given earlier (this info is in the lines 317-318). In my opinion this is the most significant bias of the study.

Authors: Only non-functioning pituitary adenomas were included. Imaging techniques for seeking parathyroid adenoma is currently recommended only in cases of suspected hyperparathyroidism and if the patient has surgical indications. This information has been included in the revised version of the manuscript

  1. Line 209: Please specify what other locations?

Authors: skin and soft tissue lesions. This information has been included in the revised version of the manuscript

  1. Line 222: What kind of cerebrovascular complications?

Authors: Stroke. The definitions of MACS-related comorbidities have been previously described in: Araujo-Castro, M., et al., Accuracy of the dexamethasone suppression test for the prediction of autonomous cortisol secretion-related comorbidities in adrenal incidentalomas. Hormones (Athens), 2021. 20(4): p. 735-744.

  1. Lines 273, 278, 325- this is confusing, do the Authors mean MACS and thyroid adenomas and other adenomas?

Authors: all types of adenomas. Specific comparisons are detailed later in the results section. For avoiding confusion, this information has been explicitly included in  the revised version of the manuscript

  1. Line 319: “other benign lesions were not identified”- other adenomas were described before (line 209).

Authors: we corrected this sentence in the revised version of our manuscript “only endocrine-related adenomas were included in the analysis and other benign lesions were not included”-

  1. Where there any cases of thyroid ca?

Authors: No, only benign lesions were evaluated in this srudy. This information is included in the Material and Method section “…only nodules with low suspicious of malignancy (using Tirads score [23]), a negative fine needle aspiration punction for malignancy or a histologically confirmed benign thyroid tumor were included in this study…”

  1. What was the size of the AI? Where all the AI benign?

Authors: only benign lesions were included in this study. This information is included in the Material and Method section “…Adrenal incidentalomas were characterized using unenhanced computed tomography (CT) scans or magnetic resonance imaging (RMI) as benign lesions…”

Round 2

Reviewer 2 Report

Please see the attached review.

Author Response

We thank the reviewer for the comments. Changes are depicted in red.

Adrenal adenoma size has been reported in table 1 and in the results section of the revised version of the manuscript. The absolute number of patients with clinical findings in additional imaging techniques is depicted in Table 1.